# Novel Clustering Methods Identified Three Caries Status-Related Clusters Based on Oral Microbiome in Thai Mother–Child Dyads

**DOI:** 10.3390/genes14030641

**Published:** 2023-03-03

**Authors:** Samantha Manning, Jin Xiao, Yihong Li, Prakaimuk Saraithong, Bruce J. Paster, George Chen, Yan Wu, Tong Tong Wu

**Affiliations:** 1Department of Biostatistics and Computational Biology, University of Rochester Medical Center, Rochester, NY 14642, USA; 2Eastman Institute for Oral Health, University of Rochester Medical Center, Rochester, NY 14642, USA; 3Master of Public Health Program, Department of Public & Ecosystem Health, Cornell University, Ithaca, NY 14853, USA; 4Department of Internal Medicine, Medical School University of Michigan, Ann Arbor, MI 48109, USA; 5Molecular Microbiology & Genetics, The Forsyth Institute, Cambridge, MA 02142, USA; 6Harvard School of Dental Medicine, Harvard University, Boston, MA 02142, USA

**Keywords:** oral microbiome, saliva, early childhood caries, clustering, maternal influence

## Abstract

Early childhood caries (ECC) is a disease that globally affects pre-school children. It is important to identify both protective and risk factors associated with this disease. This paper examined a set of saliva samples of Thai mother–child dyads and aimed to analyze how the maternal factors and oral microbiome of the dyads influence the development of ECC. However, heterogeneous latent subpopulations may exist that have different characteristics in terms of caries development. Therefore, we introduce a novel method to cluster the correlated outcomes of dependent observations while selecting influential independent variables to unearth latent groupings within this dataset and reveal their association in each group. This paper describes the discovery of three heterogeneous clusters in the dataset, each with its own unique mother–child outcome trend, as well as identifying several microbial factors that contribute to ECC. Significantly, the three identified clusters represent three typical clinical conditions in which mother–child dyads have typical (cluster 1), high–low (cluster 2), and low–high caries experiences (cluster 3) compared to the overall trend of mother–child caries status. Intriguingly, the variables identified as the driving attributes of each cluster, including specific taxa, have the potential to be used in the future as caries preventive measures.

## 1. Introduction

Early childhood caries (ECC) is one of the most common chronic childhood diseases, with nearly 1.8 billion new cases per year globally. Specifically, the risk is higher among children in Thailand; more than 50 percent of Thai 3-year-old children have experienced ECC according to the Bureau of Dental Health (2018). A prospective study demonstrated how ECC progressed rapidly among 9–18-month-old children in Southern Thailand (reported incidence of 2% at 9 months, 22.8% at 12 months, and 68.1% at 18 months of age) [1]. Another prospective cohort study in Northern Thailand found a high prevalence of severe ECC at 36 months of age (44.1%) [2]. This means that a large group of preschool children are still at risk and remain untreated. Therefore, it is a matter of urgency to enable anticipatory guidance of a caries risk assessment and the early management of dental problems.

As a multifactorial and ecology-based disease [3], the interplay between the host, environment, and oral microbiota affects the onset and severity of ECC. In recent years, maternal influences on ECC have been studied extensively, which has led to the development of conceptual frameworks [4,5] which suggest that mothers are the primary source of bacterial acquisition/colonization (vertical transmission) and socialization for their children (role nurturing) [6]. While revisiting the children’s dental caries risk model described by Fisher-Owens and others [7], several factors that are related to ECCs etiology could potentially be influenced by mothers, including: (a) biological factors such as oral microflora and genetic endowment, (b) health behavior factors such as oral hygiene practices, feeding behavior, and the utilization of medical and dental care, and (c) social environmental factors such as family socioeconomic status, parents education and health status, family function, and coping skills.

Mafla et al. investigated 384 mothers and their 2–5-year-old children in Colombia in 2020 and revealed that the number of mothers with decayed, missing, and filled tooth surfaces was positively associated with that of their children after adjustments for sociodemographic and behavioral factors [6]. A recent prospective cohort study [8], that included 1015 women and their children in Appalachia, USA, found that children whose mothers had two or more prior pregnancies, smoked cigarettes post-partum, or had a recent unfilled carious lesion were at least twice as likely to experience a dental lesion by the three-year visit. This study further indicated that untreated maternal decay was associated with a cumulative risk of ECC by age three, and the influence was modified by maternal education and state of residence. Another prospective study in Australia investigated the trajectory of the maternal intake of sugar-sweetened beverages during the first five years of their child’s life and its effect on childhood caries. This study categorized mothers’ sugar-sweetened beverages intake into three levels: “Stable low”, “Moderate but increasing”, and “High early”. This study found that the children of mothers in the ‘High early’ and ‘Moderate but increasing’ groups had greater experience with dental caries than those in the ‘Stable low’ trajectory, respectively [9]. In the context of ECC, our group assessed the maternal relatedness of oral microbiome in a set of mothers and preschool children [10]. We observed that the microbial community composition and diversity within the mother–child dyads were similar (although species abundance differed across pairs) despite the children’s caries status [10].

Despite the vast majority of data supporting the maternal influence on ECC, the factors affecting the maternal relatedness of a child’s oral health and, particularly, oral microbiome still needs to be characterized. However, there might exist heterogeneous latent subpopulations between mothers and children that have different characteristics in terms of developing caries. Since the complex causal pathways likely to be involved in maternal–child oral health relatedness have not been fully elucidated, a deeper understanding of these links will need to derive from research that has comprehensively considered multiple factors, including microbial, socio-behavior, and environmental factors.

The current study examined 177 Thai mothers and their 3-year-old biological children. Our objective was to analyze how the maternal factors and the oral microbiome of the mother and child influence caries development. A novel clustering method performed clustering and modeling with regularization simultaneously to assess the relationship between the longitudinal categorical outcomes and cluster-specific microbes in individual clusters. Our study revealed three latent clusters within the dataset, each with its own unique outcome trajectory trend, as well as identifying several microbial factors that contribute to the estimation mechanism of the clustering.

## 2. Materials and Methods

### 2.1. Study Population

This cross-sectional study was a subset of a larger study that was previously described [11,12]. From July to December of 2009, 177 3-year-old children and their biological mothers were enrolled in the current study during the children’s immunization visit at the Health Promoting Hospital in Chiang Mai, Thailand. Children with major congenital anomalies, chronic illness, or who had taken antibiotics within 6 weeks of the examinations were excluded. A structured questionnaire was used to collect data on family socio-demographics (age, sex, primary care provider, and maternal/family background), maternal pregnancy and past medical history (mode of delivery, gestational age, and birth weight), child’s feeding practice (breast feeding, bottle feeding, and mother prechewing food), dietary habits (consumption of fruit juice, snacks, gum, lollipop candy, and dried fruit), and maternal/family background. The study protocol was approved by the Ethical Committee of the Faculty of Dentistry, Chiang Mai University, Thailand. Written informed consent was obtained from all mothers or responsible caregivers at the time of the children’s hospital visits.

### 2.2. Dental Examination and Saliva Collection

Two calibrated dentists performed a comprehensive dental examination on all children and their mothers in accordance with the WHO criteria for decayed, missing, and filled teeth (DMFT for the mothers and DMFT for the children) [13]. The presence of ECC was also recorded as a detectable white-spot lesion or cavity following the American Academy of Pediatric Dentistry criteria [14]. After chewing paraffin wax for one minute under medical supervision, mothers and children provided 1 cc of whole saliva. Saliva samples were collected in centrifuge tubes and stored in the microbiology laboratory’s −20 freezer until microbiome analysis.

### 2.3. DNA Extraction and 16S rRNA Sequencing

DNA extraction and 16S rRNA sequencing were carried out at the Forsyth Institute, Harvard University. Briefly, 500 uL of saliva samples were used for whole genome DNA extraction using the Epicentre MasterPure^TM^ DNA Purification kit. A total of 129 genus-specific probes and 638 species-level probes were used for probe sequencing. QIIME 1.9.1 [15] was used to quantify the composition and diversity of each community based on its open-reference OTU picking facility. Sequencing data that passed quality controls were included in this study to develop a caries predictive model and were assigned to operational taxonomic units (OTUs). OTUs which have zero counts across all the samples, or only appear in one sample, were removed from the further analysis (see Appendix A for the sequence sample size by the mother–child groups). The sequence reads of all the samples in the study are deposited in the NCBI Sequence Read Archive (SRA) as a study under the accession number of PRJNA824062.

A total of 253 low abundance (less than 4 count, and present among less than 20% of all samples) features were removed based on prevalence. A total of 21 low variance features were removed; the number of features which remained after the data filtering step was 184. The phyloseq package was used to analyze the α diversity. For each group or experimental factor, the results were plotted across samples and reviewed as box plots. In addition, the statistical significance of grouping based on the experimental factor was estimated using the t-test. The α and β diversity similarity or distance between the samples was calculated using non-phylogenetic distances (Shannon index and Bray–Curtis distance). Principle coordinate analysis (PCoA) was used to visualize these matrices in a 2D plot, with each point representing the entire microbiome of a single sample. Permutational ANOVA was used to determine the statistical significance of the clustering pattern in ordination plots (PERMANOVA). LDA effect size (LEfSe) analysis employs a non-parametric factorial Kruskal–Wallis (KW) sum-rank test to identify features with a significant differential abundance in relation to the experimental factor or class of interest, followed by linear discriminant analysis (LDA) to compute the effect size of each differentially abundant feature. Features are considered to be significant based on whether they have an adjusted *p*-value less than 0.05.

### 2.4. Clustering Methods

This model-based clustering method was built for the clustering of correlated data with high dimensional covariates. The motivating idea behind it was to be able to analyze non-continuous response trajectories from longitudinal or dependent observations. It is important to be able to identify latent heterogenous subpopulations in these sort of data to better understand the relationship between the independent and response variables. In this paper, we aimed to cluster the mother–child dyads based on their caries patterns. Differing patterns, or latent groupings, may indicate the need for different treatment plans, which would be useful for clinicians to be able to identify and assign. The clustering algorithm has two prominent alternating stages after the cluster initialization.

First, a generalized linear mixture model with LASSO penalization (GLMMLASSO) was fitted within each cluster. This was done using the glmmLasso package in R [16]. Random intercepts were included in the GLMMLASSO model to account for the correlation of between mother and child for each dyad. The logit link function was used since the outcome variable of interest indicates the presence or absence of caries (caries = 1, no caries = 0), a binary variable, which was collected from both mother and child. To obtain unbiased and consistent estimates, the nonzero coefficients of said models were then re-estimated using the glmer function of the lme4 library in R without any penalty using data in the cluster. This re-estimation helped to reduce the bias caused by the LASSO penalization. The generalized linear mixed-effects model with re-estimated coefficients was then used to calculate the probabilities of having caries for both mother and child in each dyad in the corresponding cluster. To choose the optimal value of the penalty parameter lambda, which determined the number of selected variables in the GLMMLASSO model, a unique value was selected in each iteration in the model fitting procedure for each cluster. The optimal value of lambda was chosen using the Akaike information criterion (AIC), calculated from each model in each iteration. This allowed for a greater flexibility due to the changing cluster size and composition each of the models were subject to.

The second stage of the algorithm is the clustering procedure. The estimated probabilities were used as the input of a k-means procedure for longitudinal response trajectories [17], with k = the number of clusters. This essentially treats the correlated observations of each dyad/trajectory (in this case, the estimated probabilities of both mother and child) as the two measures of interest, and groups the dyads/trajectories (in this case, family units) by a Euclidean measure of distance. This means that mother–child dyads with similar probabilities were grouped together, as we are interested in the influence of the selected variables on the probabilities of mothers and their children in respect to developing caries. This new clustering assignment was then used to repeat the first stage of the algorithm. The two stages were alternated between until the cluster membership stopped changing between the two iterations.

For this implementation, the number of clusters was determined to be three, and the clusters were initialized using a k-means procedure on the independent variables of the data, treating each mother–child dyad as one observation. Note that this initial estimation does not take into account the outcome variable.

## 3. Results

The algorithm took 29 iterations to reach cluster convergence. Table 1 shows the key summary statistics in regard to the final cluster assignments. The first cluster had caries statistics which match closely to the overall average for the dataset. Here, we could call these the “typical” mother–child dyads, as they showed no major divergence from the overall set. Cluster 2 was particularly intriguing because the mothers in this cluster were more likely to have caries, whereas their children had the lowest caries rate among the three clusters. We named this cluster the “high–low” group. Cluster 3 also had an interesting composition. The mothers in this cluster had a lower caries rate compared to the overall caries rates of mother in the entire dataset, while their children’s caries rate was higher than the overall average and was the highest among the three clusters. We named this cluster the “low–high” cluster. Both “high–low” and “low–high” clusters are worth further investigations as they show the opposite maternal relatedness and contain insight into the prevention of caries.

Table 2 contains the cluster-specific coefficients. These coefficients can be interpreted as important factors associated with developing caries in the corresponding cluster. Specifically, a covariate with a positive sign would indicate that the presence variable in question indicates the higher odds of a subject developing caries in that cluster. A negative sign would indicate the opposite. Although only microbial factors were identified by the clustering method, the post-clustering analysis revealed additional variables that show differences between clusters.

We can interpret the coefficients more contextually as follows. For example, four microbes were identified in cluster 3 (the “low–high” cluster), meaning that the four microbes are significantly related to the caries status in this cluster. The positive regression coefficients indicate possible infection-inducing pathogens. The variable *O. sinus* had a coefficient value of 0.417 in the generalized linear mixed-effects model using the logit link, thus the odds ratio of possessing caries is approximately 1.52, controlling for other variables and ignoring the random intercepts. This would indicate that a one-unit increase in the relative abundance of *O. sinus* would increase the predicted odds, and hence the probability, of possessing caries.

Following cluster assignment, we analyzed the diversity, taxa relative abundance, and differentially abundant taxa at the genera level between the mothers and their children among the three clusters. The α diversity measured by the Shannon index was only significantly different between the cluster 2 mothers and children with less experience of caries (Figure 1A,B). Regarding the β diversity, mothers and children were significantly different in cluster 1 and cluster 2 (Figure 1C–E). In cluster 1, where mothers and children share similar caries experiences, mothers harbor a higher amount of *Haemophilus*, *Neisseria*, and *Streptococcus*, but less *Veillonella*, Rothia, Prevotella, and *Leptotrichia*, etc., than their children (Figure 1F). In cluster 2, where the disparity between mothers with caries and children with caries was greater, mothers had a higher abundance of *Neisseria* and *Aggregatibacter,* and children were enriched with a higher abundance of Actinomyces, Capnocytophaga, Saccharibacteria, Rothia, Prevotella, and Leptotrichia (Figure 1G). In cluster 3, where children had more caries experience (64.3%), the children’s saliva was more enriched with *Rumiococcaceae* and *Alloprevotella* (Figure 1H).

Furthermore, post-clustering analysis was performed. First, we examined the factor-level variables of interest to see if any were significantly different between the clusters. To ensure that the false discovery rate (FDR) was controlled, the Benjamini–Hochberg procedure was performed on all *p*-values of the comparisons. Figure 2 contains bar plots for the significantly different maternal factors across clusters in addition to an indicator of their *p*-value range after adjustment.

Moreover, the selected OTUs were also checked to see if they were significantly different between the clusters (Figure 3). As there were many OTUs in the data, it would be imprudent to check all of them, hence why the comparison was limited to those that were selected. Again, the Benjamini–Hochberg procedure was performed on all *p*-values to control the FDR. The distributions for each cluster of the significantly different selected OTUs are contained in Figure 3 in addition to an indicator for their *p*-value ranges.

## 4. Discussion

There are two popular approaches that are used to cluster microbiome data. Both approaches conduct clustering and modeling in separate steps [12]. In the first approach, the microbiome data are grouped into different clusters on its own, disregarding the outcome and other factors, and upon obtaining cluster assignments, attempt to interpret them in reference to the outcome of interest [18]. The initial clustering can be conducted in any number of ways, with varying success, but the interpretation can be somewhat lacking in rigor. The second approach is latent class analysis/regression, in which the response variables are first clustered without involving any covariates and then the cluster membership is used as the response in a multinomial regression model with the covariates of interest to attempt to unearth any connections between the two [12]. Both approaches require two separate steps in order to fully connect the outcome and covariates of interest to the cluster assignments. It is obvious that these other approaches fail to consider the relationship between the covariates and the outcome of interest during the clustering process as they only make such a connection after the clustering has been performed. In addition, limited work has been done on the clustering of longitudinal high-dimensional data [19]. The longitudinal latent class analysis may be used to consider the correlation inherent in time-dependent, repeated-measure observations, with the limitation that all time points must be identical across subjects.

In this paper, we consider the clustering of correlated observations with categorical outcomes and high-dimensional microbiome data. The method used here is a novel, non-trivial extension of the one detailed in Yang and Wu [19], which is a mixture-model-based clustering method for longitudinal data with regularization to enforce a variable selection of high-dimensional covariates. We extend this by considering categorical outcomes as opposed to Gaussian outcomes, which pose unique challenges of their own, as the Yang and Wu method only considers Gaussian outcomes. In particular, we focus on correlated binary outcomes in this paper. This novel clustering method performs clustering and modeling with regularization simultaneously to assess the relationship between the longitudinal categorical outcomes and cluster-specific microbes in individual clusters.

Our method improves upon that in the following ways: first, we consider the potentially crucial relationships while determining cluster membership; second, our method conducts clustering and modeling simultaneously, so after the algorithm converges, the results contain both the cluster membership and covariate relationships; and third, our method performs variable selection for each cluster in the model fitting process, i.e., each cluster has its own cluster-specific covariates that explain the relationship with the outcome. The method is designed for nested data, and as such, it can be applied to both repeated measures data (longitudinal data) or correlated data (as in this paper). There is no requirement of any equanimity in the time of or number of observations, which makes it more flexible than traditional longitudinal latent class analysis. As of now, it is important to note that no other current clustering method specifically accounts for correlated data, high-dimensional covariates, and non-Gaussian outcomes in one procedure.

It is important to develop this sort of analysis as the current oral health community does not necessarily consider the possibility of the existence of latent subpopulations within patients. Heterogenous subpopulations will not necessarily respond the same way to the same treatment plan, especially so when considering the whole of oral health and its possible conditions. The different clusters, as further described below, could indicate the need for different treatment plans. If it is possible to assign new patients to these groupings, we could optimize their treatment plan and improve their overall health. A further investigation is needed to confirm these clusters and their identified factors as this is the first instance of the identification of these latent groupings.

The clusters determined by the described method provide new insight into the relationship between the oral microbiome, maternal factors, familial relatedness, and the development of ECC. In the third cluster, the number of mothers with caries (85.71%) was quite low compared to the overall incidence (93.71%). In contrast, the number of children in that cluster with caries was higher than the overall rate by a good margin (64.29% as compared to 53.14%). This cluster should contain an insight into how a disease-free, or at least a lower-risk mother, might still have a child with ECC, possibly indicative of the causal factors of ECC. The second cluster is the opposite: the mothers were more likely than average to possess caries, whereas their children were less likely. This cluster may point to protective factors in children, and possible avenues to prevent ECC in children despite ill oral health in the mother. The first cluster appears to be closer to the average in the mother’s caries incidence, with children having a higher incidence rate than average. As the average for the mother’s caries incidence is still quite high, this too could be an indicator for causal factors of ECC, though in this case, without the propensity for healthier mothers as in cluster 3. A further study is required to confirm these findings, but the unearthing of latent groupings of these mother–child dyads could be an indicator of the different pathways of development, or lack thereof, of ECC in the children.

Oral mucosal surface microbial colonization begins at birth with the introduction of bacteria and fungi via a variety of routes, including maternal perineum-infant oral transmission during childbirth [20], parental environmental exposures, digit sucking, diet, and horizontal transmission from caregivers and peers [21,22,23]. Previous studies suggested that the transient maternal transmission of strains from multiple sources, such as vaginal, skin, oral, and gut communities, all contribute to the early colonization of infant microbiomes [24]. The study from Mason et al. further revealed that 85% of infants’ oral microbiota resemble their mothers in early life (6 months of age), but the microbiota differs with respect to tooth eruption in the children’s mouths at around 6-to-8 months of age [21]. A review of the literature in this field agrees with this conclusion, noting that infant microbial diversity is significantly lower than in their parents, but maintains that there is a significant association between the mother’s oral microbiome and her infant’s [25]. Interestingly, another review notes that children with a higher heterogeneity in their oral microbiomes have lower rates of caries incidence [26].

Similar to gut microbial development in early infancy, some studies suggest that the delivery modes—vaginal delivery or a cesarean section—also influence infants’ oral microbial community development; however, the term of influence remains unclear. Previous studies reported that *Lactobacillus*, *Prevotella*, *Bacteroids*, *TM7*, and *Sneathia* species were the predominant bacteria in vaginally delivered infants’ saliva. In contrast, the bacterial profile of infants born by cesarean section resembles those present in their mothers’ skin, predominantly *Staphylococcus*, *Corynebacterium*, *Slackia*, *Veillonella,* and *Propionibacterium* spp. [20,27,28]. Interestingly, this study identified the factor of vaginal delivery as significantly different between the three clusters, which could be a potential factor associated with the trajectory of caries experience.

A limitation of this cross-sectional designed study is that the study cannot assess the prospective effect, as in the future possible oral developments of the child. More well-designed longitudinal birth studies are needed to confirm these projections. Furthermore, this study used a probe sequencing tool that yields species-level resolution instead of strain-level comparison between mothers and children. This study’s limitation could be overcome by taking advantage of the strain-level resolution of metagenomic sequencing methods, such as the metagenomic SourceTracker approach [29], to further determine the maternal origins of infant oral microbial samples.

## 5. Conclusions

Using the described novel clustering technique, three clusters were identified within the data, each with their own differing trajectory pattern for the response variable of the presence or absence of dental caries. Intriguingly, the responsible variables identified as the driving attributes of each cluster, including specific taxa, have the potential to be used as caries causal and preventive factors in children and their mothers.

## Figures and Tables

**Figure 1 genes-14-00641-f001:**
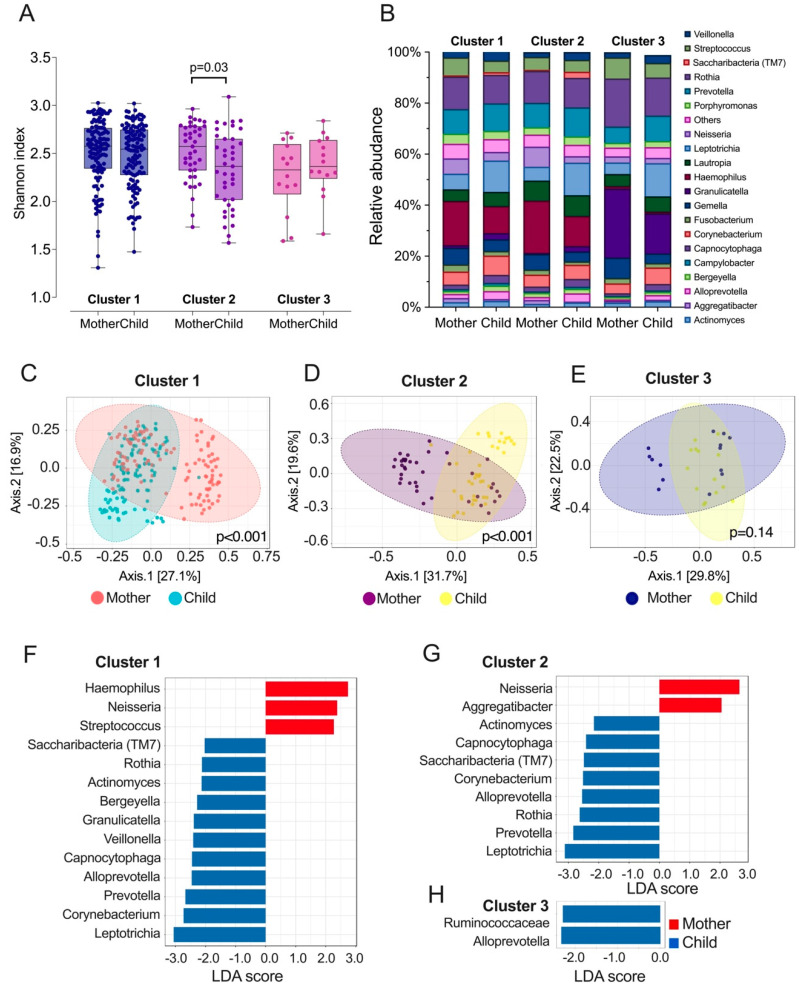
Salivary l microbiome profile of the three clusters identified among Thai mother–child dyads. (**A**) α diversity of salivary microbiome among mothers and children. Microbial variation measured by α diversity Shannon index. *t*-test was used for statistical analysis. (**B**) Relative abundance of top 20 genera among mother and children. (**C**–**E**) β diversity of salivary microbiome among three clusters of mothers and children. Principle coordinate analysis (PCOA) plot is generated using OTU metrics based on Bray–Curtis index. Permutational MANOVA (PERMANOVA) was used for statistical analysis. (**F**–**H**) Taxa at genus level differently enriched in the saliva of mother and children. Linear discriminant analysis (LDA) effect size method was performed to compare taxa between children and mothers. The bar plot lists the significantly differential taxa based on effect size [LDA score (log10) > 2 and FDR < 0.05].

**Figure 2 genes-14-00641-f002:**
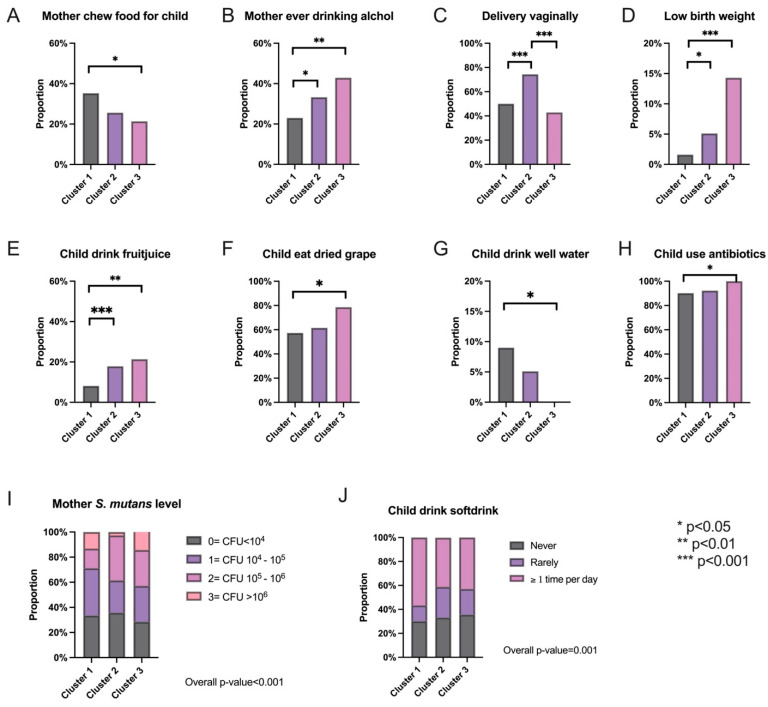
Significantly different maternal factors identified between the three clusters. (**A**–**H**) Bar plots of the proportion of subjects within each cluster that indicated yes to the labeled maternal factor of interest. The ones shown above are the ones that are significantly different between the clusters. (**I**,**J**) Proportions of each level of the indicated maternal factor within each cluster.

**Figure 3 genes-14-00641-f003:**
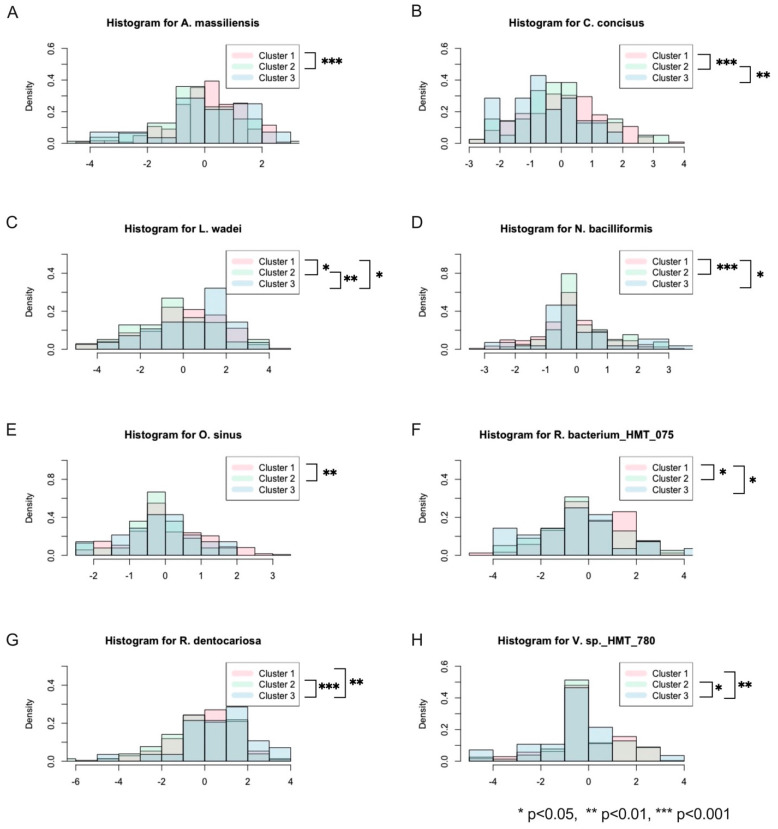
Histograms of significantly different selected OTUs across the three clusters. (**A**–**H**) The density of the abundance each OTU within each cluster is represented in a layered fashion to visualize the differences between them.

**Table 1 genes-14-00641-t001:** Clusters generated by novel GLMM-LASSO clustering method.

	Both with Caries	Mothers with Caries	Children with Caries	Neither with Caries	Total with Caries	Cluster Size
Cluster 1 (typical)	54.10%	93.44%	57.38%	3.28%	75.41%	122
Cluster 2 (high–low)	48.72%	97.44%	48.72%	2.56%	73.08%	39
Cluster 3 (low–high)	57.14%	85.71%	64.29%	7.14%	75.00%	14
Overall	56.00%	93.71%	53.14%	3.43%	74.85%	175

**Table 2 genes-14-00641-t002:** Estimated coefficients from the cluster-specific GLMM-LASSO models.

Cluster 1		Cluster 2		Cluster 3	
Species	Coefficients	Species	Coefficients	Species	Coefficients
*Prevotella oulorum*	6.99	*Alloprevotella tannerae*	0.14	*Actinomyces massiliensis*	0.22
		*Campylobacter concisus*	0.04	*Peptostreptococcaceae yurii*	0.09
		*Capnocytophaga* sp.*_HMT_903*	0.31	*Oribacterium sinus*	0.42
		*Ruminococcaceae bacterium_HMT_075*	0.26	*Veillonella* sp.*_HMT_780*	0.10
		*Filifactor alocis*	0.005		
		*Gracilibacteria bacterium_HMT_872*	0.05		
		*Leptotrichia wadei*	0.02		
		*Neisseria bacilliformis*	0.08		
		*Porphyromonas _HMT_279.2*	−0.02		
		*Prevotella denticola*	0.05		
		*Rothia dentocariosa*	0.08		
		*Streptococcus sanguinis*	0.05		
		*Streptococcus* sp.*_HMT_431*	0.16		
		*Saccharibacteria bacterium_HMT_348*	0.04		

## Data Availability

The sequence reads of all samples in the study are deposited in the NCBI Sequence Read Archive (SRA) as a study under the accession number of PRJNA824062.

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
