# Peer review of "Novel Clustering Methods Identified Three Caries Status-Related Clusters Based on Oral Microbiome in Thai Mother–Child Dyads"

_genes, 2023, doi:10.3390/genes14030641_

Round 1

Reviewer 1 Report

 The paper introduce a novel method to cluster the oral microbiome in Thai mother-child dyads. I congrats with the author for the well written methodology and for the novelty of the results. However, in the discussion I strongly suggest the integration of review studies analyzing what happens in healthy children and infants, please add the following citations:

https://pubmed.ncbi.nlm.nih.gov/36141674/

https://pubmed.ncbi.nlm.nih.gov/33441592/

Author Response

We thank the reviewer for their kind words, and for their helpful suggestion. The requested citations have been added to the end of the sixth paragraph in the discussion.

Reviewer 2 Report

Samantha Manning et al.,  adequately described the new cluster based on oral microbiome in Thai Mother-Child dyad. The novel new method of clustering was also described in details. Overall the manuscript is well written with adequate references.

1) Did the authors compared the cluster analysis with Yang and Wu method? 

2) How this clustering analysis is better than any other methods and what are the advantage using this technique.

Author Response

We thank the reviewer for their helpful and insightful comments. All the suggested changes have been made.

1) Did the authors compared the cluster analysis with Yang and Wu method? 

We thank the reviewer for pointing out that this was unclear. A sentence addressing why this comparison was impossible was added to the second paragraph in the discussion.

2) How this clustering analysis is better than any other methods and what are the advantage using this technique.

Thanks for emphasizing this point. We have addressed this idea at the end of the third paragraph in the discussion.

Reviewer 3 Report

In this work, the authors have examined a set of saliva samples of Thai mother-child dyads and have analyzed how the maternal factors and oral microbiome of the dyads influence the development of early childhood caries. The authors have developed three heterogeneous clusters in the dataset, each with its own unique mother-child outcome trend, as well as identifying several microbial factors that contribute to early childhood caries.

It is a study that also takes into account different independent variables that may have an influence on the development of caries. It is a complete work.

Author Response

We thank the reviewer for their kind words. No changes have been made in accordance with their review.